# *Opuntia ficus-indica* Fruit: A Systematic Review of Its Phytochemicals and Pharmacological Activities

**DOI:** 10.3390/plants12030543

**Published:** 2023-01-25

**Authors:** Luis Giraldo-Silva, Bárbara Ferreira, Eduardo Rosa, Alberto C. P. Dias

**Affiliations:** 1Centre of Molecular and Environmental Biology (CBMA), University of Minho, Campus de Gualtar, 4710-057 Braga, Portugal; 2Centre for the Research and Technology of Agro-Environmental and Biological Sciences (CITAB), University of Trás-os-Montes and Alto Douro, 5000-801 Vila Real, Portugal

**Keywords:** prickly pear, antioxidants, betalains, phenolics, bioactive compounds, food and health

## Abstract

The use of *Opuntia ficus-indica* fruits in the agro-food sector is increasing for a multiplicity of players. This renewed interest is, in part, due to its organoleptic characteristics, nutritional value and health benefits. Furthermore, industries from different sectors intend to make use of its vast array of metabolites to be used in different fields. This trend represents an economic growth opportunity for several partners who could find new opportunities exploring non-conventional fruits, and such is the case for *Opuntia ficus-indica*. *O. ficus-indica* originates from Mexico, belongs to the Cactaceae family and is commonly known as opuntia, prickly pear or cactus pear. The species produces flowers, cladodes and fruits that are consumed either in raw or in processed products. Recent publications described that consumption of the fruit improves human health, exhibiting antioxidant activity and other relevant pharmacological activities through enzymatic and non-enzymatic mechanisms. Thus, we provide a systematic, scientific and rational review for researchers, consumers and other relevant stakeholders regarding the chemical composition and biological activities of *O. ficus-indica* fruits.

## 1. Introduction

*Opuntia ficus-indica* (L.) Mill, commonly known as prickly pear, Indian fig or nopal is a crassulacean acid metabolism (CAM) plant belonging to the Cactaceae family, *Opuntia* genus. Originating from Mexico, it is widespread through Central and South America, Australia, the Mediterranean basin and South Africa [1]. It is believed that *O. ficus-indica* was first introduced in Africa and Europe because it is the host of the cochineal insect *Dactylopius coccus*, a source of a commercial red dye [2]. The species can grow in arid and semiarid regions at high temperatures and low water availability [3].

Fresh cladodes, fruits and flowers are traditionally used for different purposes. Cladodes are rich in fibers such pectin, lignin, cellulose and hemicellulose [4] and can be used as animal feed, fodder or for human consumption [5]. Flowers are normally used as infusions for their diuretic activity [1].

*O. ficus-indica* fruits can present large color differences among cultivars, varying from green to white, yellow to orange and red to purple. These variations can be attributed to the betalain type pigments [6,7,8,9]. Fruits are highly flavored and a source of interesting and useful compounds, such as polyphenols [10,11,12,13,14,15,16], dietary fibers [17,18,19], carotenoids [9,20] vitamins [21], minerals [22] and amino acids [23,24]. *O. ficus-indica* fruit demand is increasing in domestic and international markets due to growing recognition of its nutritional and health value [25,26]. In addition, the species represents an opportunity for local growers to gain access to superior markets in which consumers place emphasis on exotic character and quality traits. 

*O. ficus-indica* fruit or juice consumption may exert antioxidant activity through non-enzymatic mechanisms [27] or modifying SOD, CAT and GSH enzymatic levels [28]. Polyphenols present in syrup concentrates can display anti-cancer activity in tumorigenic lines of fibroblasts and neuroblastoma [11], and fermented juice could reduce UV-B damage induced in fibroblasts [29]. Besides its pharmacological importance, fruits have a considerable role contributing to the individual daily intake of minerals and other essential nutrients when are consumed fresh [22] or as food supplements [30].

The aim of this paper is to highlight the scientific literature and respective knowledge about *O. ficus-indica* fruit, concerning their biological activities and chemical composition relevant for food consumption, providing a systematic, scientific and rational review for consumers, stockholders and researchers.

## 2. Results and Discussion

### 2.1. Literature Search

In this review, the literature search using “*Opuntia. ficus-indica*” and “prickly pear” keywords identified the following number of articles in the respective databases: PubMed^®^ (*n*= 1369), ScienceDirect^®^ (*n* = 2331) and Scopus^®^ (*n* = 3335). After removing duplicates and excluding papers according to the methodology depicted in Figure 1, 482 articles were included by title, and from these, 367 articles were included by abstract and only 194 articles were selected by full text availability and were finally included in this study (Figure 1). 

Since a single document sometimes delt with several topics, it is important to highlight that some of the analyzed papers were classified more than once into different categories. This resulted in a larger number of items, defined as “records”. In addition, once the total number of records assigned to the main categories was analyzed, it was found that some records could, in turn, be reclassified into subcategories, increasing again the number of items and were ultimately defined as “sub-records”. The search strategy is summarized in Figure 1.

After the analysis of the 194 selected documents, a total number of 369 records was obtained, which were organized in three (3) categories: (i) phytochemicals, (ii) betalains and (iii) biological activities (Figure 2). 

In the first approach, the betalains were included in the phytochemicals category. However, due to their specific characteristics and due to the nature of the information output, e.g., extraction procedures, quantification, identification by analytical techniques, thermal stability and potential use in medicine and other industries, a single and unique category was created for these compounds.

#### 2.1.1. Phytochemicals

After screening the whole documents, it was found that 35.7% of the total information was related to the chemical characterization and composition of *O. ficus-indica* fruits (a total of 132 records). Once the 132 records assigned to chemical characterization were analyzed, new sub-categories were created, constituted by 223 final sub-records (Figure 3).

##### Phenolic Content

Phenolic compounds have gained special attention due to their ability to scavenge free radicals or donate hydrogens to counterpoise reactive species. Furthermore, some polyphenols may be involved in several biological process such as nitric oxide regulation, anti-inflammatory processes or may possess antimicrobial activity [31].

*O. ficus-indica* fruits phenolic composition differs qualitatively and quantitatively depending on the variety [32,33]; part of the fruit, either peel, pulp or whole fruit [11,34]; sample preparation and extraction process [35,36,37]; geographic location [9,38]; season [39]; and storage conditions [27,39,40]. A full profile of identified phenolic compounds by different techniques is summarized in Appendix A. 

In a recent study, the varieties “Colorada”, “Fresa”, “Blanco Buenavista” and “Blanco Fasnia” collected in the Canary Islands were subject to HPLC-ESI-QTOF. Among the findings, three phenolic acids and 14 flavonoids, mainly isorhamnetin glycoside derivatives, kaempferol glycosyl-rhamnoside and rutin were identified. Pulp fruit phenolic compounds ranged from 38 to 62 mg/100 g f.w for all the varieties. Peel phenolic content is significantly higher than pulp. The lowest content was reported for “Blanco Fasnia”, with 327 mg/100 g f.w, whereas the remaining varieties ranged from 4.3 to 4.5 g/100 g f.w [41]. Data about the range of total phenolic concentration (TP) based on *O. ficus-indica* fruits are presented in (Table 1).

Processing techniques have a direct impact on TP yield [20,63]. Oven dried Morocco peel TP values ranged from 1.7 to 2.4 g GA/100 g d.w depending on the variety [51]. Mexican peel and pulp fruit subjected to high hydrostatic pressure (HHP) accounted for 11 and 4 g GA/g d.w [64]. Mexican green and red fruit residues mixed either with microcrystalline cellulose or lactose showed TP values ranging from 648 to 734 mg GA/100 g d.w [65]. Australian prickly pears treated with four different drying techniques (freeze dryer, microwave, oven dehydrator at 35 and 55 °C) showed that the freeze-drying technique reached the highest TP levels [35]. Fruits grown in drainage sediment had higher total phenolic compounds values than those grown in normal conditions [66], and wounded fruit pulp irradiated with UVB for 15, 90 and 180 min increased total phenolic accumulation by 52.5%, 101.8% and 38.8%, respectively, when compared with non-treated samples [67].

##### Organic Acids

Organic acids are low weight molecular compounds directly produced from different organisms’ normal metabolism [68,69,70] and may be influenced by biotic and abiotic factors [27,71]. For instance, ascorbic acid is a water-soluble vitamin that is highly appreciated as a food additive for its antioxidant activity and is an important intermediate in redox reactions involved in bones, blood vessels and skin preservation [72]. Malic acid is normally used in candies, food and beverages as a pH control agent, flavoring agent, acidifier and preservative [73] and its consumption can improve skin conditions, boost immunity reduce risk of metal poisoning and promote oral health [74].

Organic acids accumulated in *O. ficus-indica* fruits are influenced by geographical location and processing techniques. For example, peel and pulp extracts of Portuguese varieties “Gialla”, “Rossa”, “Orange” and “Red” showed the presence of glutaric, malic, succinic and pyruvic acid [71], whereas malic, quinic, citric, piscidic acid and derivates, as well as caffeic and hydroxybenzoic acid, were identified by ESI-MS/MS from Portuguese fruit juice [75]. Sicilian “Rose”, “Gialla” and “Bianca” cultivars showed traces of succinic, lactic, pyruvic and isobutyric acid [76]. 

Ascorbic acid has been quantified in many *O. ficus-indica* fruits (Table 2). Its content seems to be higher than other organic acids [77] and also higher than the values reported for apples, grapes, bananas and pears [78]. 

##### Carotenoids

Carotenoids are considered relevant in the prevention of certain oxidative stress-related diseases such as cancer. The most studied carotenoids are β-carotene, lycopene, lutein and zeaxanthin. The first two belong to the carotene class, featured by their carbon–hydrogen structure; the second group is called xanthophylls, which, besides their carbon–hydrogen structure, also present oxygen substituents [86].

Recently, carotenoid identification and quantification assisted by HPLC-DAD/MS was carried out in saponified and non-saponified extracts from Spanish *O. ficus-indica* fruits of “Sanguinos” and “Verdal” cultivars [9]. A summary of carotenoid concentrations is presented in (Table 3). 

##### Lipid and Volatile Components

Fatty acids play pivotal roles in living organisms, being modulators of physiological functions and a source of energy [87]. Normally classified as saturated acids, monounsaturated acids (MUFAs), and polyunsaturated acids (PUFAs), their consumption has been reported to bring health benefits. For instance, linoleic and α-linolenic acid are two of the main representative compounds that cannot be synthesized by humans, known as dietary essential fatty acids, since they prevent deficiency symptoms. Additionally, PUFAs interact with nuclear receptor proteins that bind to certain regions of DNA and thereby can modulate the transcription of regulatory genes [88].

**Table 3 plants-12-00543-t003:** Carotenoid content of *O. ficus indica* fruits and juices.

Origin	Description	Total Carotenoids	Reference
**Algeria**	Orange fruit pulp	503 μg/L	[79]
**Algeria**	Orange fruit pulp	108.8 μg/100 g f.w	[42]
**Italy**	“*Sulfarina*”	1.48 μg/100 g f.w	[89]
	“*Sanguinos*”	3.47 μg/100 g f.w	
	“*Muscareda*”	1.45 μg/100 g f.w	
**Mexico**	“*Naranjona*”	85 mg /100 g f.w	[90]
	“*Blanca Cristalina*”	400 μg/100 g f.w	
	“*Esmeralda*”	700 μg/100 g f.w	
**Morocco**	Fruit juice	20.8 μg/L	[82]
**Morocco**	“*Akria*” Fruit pulp	110 μg/100 g f.w	[84]
	“*Drbana*” Fruit pulp	121.1 μg/100 g f.w	
	“*Mlez*” Fruit pulp	150 μg/100 g f.w	
**Spain**	“*Sanguinos*” Whole fruit	478.1 μg/100 g f.w	[9]
	“*Sanguinos*” Pulp	255.9 μg/100 g f.w	
	“*Sanguinos*” Peel	1.69 mg/100 g f.w	
	“*Verdal*” Whole Fruit	444.9 μg/100 g f.w	
	“*Verdal*” Pulp	379.4 μg/100 g f.w	
**Turkey**	Whole fruit Fresh	1.2 mg/100 g d.w	[91]
	Whole Fruit Frozen	1.2 mg/100 g d.w	
	Whole Fruit Sun-dried	454 μg/100 g d.w	
	Whole Fruit Microwave-dried	554 μg/100 g d.w	
**USA**	Green Skinned	290 μg/100 g f.w	[14]

Fruits collected from different farms in Alicante (Spain) were analyzed for fatty acid composition. Pulp fruit showed approximately 16.9–34.8% monounsaturated fatty acids (MUFA) and 35.2–53.9% polyunsaturated fatty acids (PUFA), with oleic and linoleic acid as major compounds, whereas peel fruit showed 11.2–31% MUFA and 37–61.1% PUFA, with linoleic, oleic and palmitic acid found in higher amounts [92]. In addition, Egyptian peel fruit contained 41.2% PUFA, 30.1% MUFA and 28.7 SFA, and eicosadienoic acid was found as the major proportion, followed by oleic and palmitic acid [46]. On the other hand, peels from Tunisian fruits showed a high amount of PUFA, 60.2%, and FA, 23.4%, while pulp and seeds showed 25 and 22.4% PUFA. The general profile showed that linoleic acid was the most abundant, followed by oleic and palmitic acid [62].

Petroleum ether saponified extract of air-dried powdered of Egyptian fruit peels exhibited 15 fatty acids, amounting to 93.2% of the saponifiable matter, where ethyl linoleate was found as the major proportion, followed by ethyl oleate and methyl palmitate. The unsaponifiable extract was composed of 33 compounds amounting for 79.2%, in which two fatty alcohols were identified as major compounds, 1-tricosanol and 1-docosanol [93]. 

The total lipid content of *O. ficus-indica* peel fruit purchased on a German market may reach up to 36.8 g/kg d.w, where 63.3% of total lipids correspond to neutral lipids, 26.6% to glycolipids, and 8.75% to phospholipids. In addition, 14 fatty acids were identified ranging between C12 and C26, distributed across total lipids and neutral lipid fractions [21]. Insight suggests that recovered lipids could be suitable for commercial exploitation for food use or production of cosmetics.

The volatile composition may be affected by growing location, e.g., Italian and Greek juices of *Opuntia* fruits grown on different locations showed different volatile profiles, normally dominated by alcohols, aldehydes and, in minor proportion, terpenoids, which were the most abundant in all samples [94]. 

The extraction procedure has a direct effect on volatile profile, e.g., peel treated by ultrasonic-assisted solvent extraction showed different a profile than peels subjected to Soxhlet. This can be explained by the high temperature used in Soxhlet extraction that might be degraded in some compounds such as phytol, azulene and serverogenin acetate [36]. 

Headspace solid-phase microextraction (HS-SPME) and headspace GC-MS were used to analyze and summarize the volatile component of red, white and yellow Sicilian varieties. The most abundant compounds for all varieties were alcohols and esters, followed by terpenes and aldehydes [95]. The volatile profile of six Spanish varieties analyzed by HS-SPME followed by GC-MS resulted in the identification of 35 compounds, 28.5% were aldehydes, 20% terpenes, 20% esters, 17.1% alcohols, 5.7% terpenoids, 2.8% ketones and 2.8% linear hydrocarbons. The only common compounds in all varieties were ß-myrcene, p-cymene, D-limonene, (E)-ß-ocimene, y-terpinene, linalool, nonanal and 2,6-nonaadienal [96]. 

Sixteen volatile compounds were identified from yellow, red and white cultivars, using HS-SPME followed by GC-FID, and complemented by GC-MS when necessary. Only (E)-2-hexanal, (Z)-2-penten-1-ol and (E,Z)-2.6-nonadien-1-ol were found in all analyzed samples. The total range of volatile compounds was 8 mg/kg for red fruits, 11 mg/kg for yellow fruits and 10.8 mg/kg for white [97]. Furthermore, the n-hexane fraction of pinkish collected in Turkey resulted in fourteen compounds in which the main components were hexadecanoic acid (39.4%), followed by heptacosane (12.3%), methyl linoleate (6.8%), hexacosane (5.8%), tricosane (5.1%), methyl hexadecanoate (4.2%), camphor (2.8%), borneol (2.5%), verbenone (1.8%), pentacosane (1.7%) and α-terpineol (1.1%) [98]. 

Over 40 volatile compounds were identified as primary aldehydes and acids, however, results suggest that volatiles cannot be used as a fingerprint for sensory differentiation among cultivars [76].

#### 2.1.2. Betalains

Betalains have developed growing interest as natural dyes due to their lack of toxicity, low cost, friendly extraction technology and easy application as food additives or nutraceuticals. Betalains are water-soluble pigments manly restricted to the Caryophyllales order and some fungi species from Basidiomycetes [99]. They are subdivided into two main types, red-purple betacyanin and yellow-withe betaxanthins [100], both considered to have antioxidant potential [20]. 

After document screening, 22% of the total information discussed is related to *O. ficus-indica* betalain (81 records) (Figure 2). This information was further divided into sub-categories and the final number of accounted new sub-records for this category was 120, where 62.5% of the information is related to pigment quantification. We also found that 65.3% of pigment quantification information is carried out by spectrophotometric techniques and the other 34.6% is usually performed assisted by chromatographic methodologies (Figure 4B). 

Betalain identification is mainly performed by mass spectrometry and corresponds to 12.5% of collected information, followed by information about pigment stability, with 12.5%, and finally 2.5% was related to metabolic pathway information (Figure 4).

Identification of *O. ficus-indica* betalains have been performed several times [16,99,101]. Betalain profile of pulp and peels obtained from three different Portuguese fruits showed seven compounds, two isomers of indicaxanthin and five betacyanins (betanidin-5-O-β-sophoroside, etanidin-5-O-β-glucoside (betanin), isobetanin, gomphrenin I and betanidin) [70]. Cultivars “Naranjona” and “Roja pelota” subjected to HPLC-ESI-MS showed the presence of portulacaxanthin I, vulgaxanthin i, vulgaxanthin ii, vulgaxanthin iii, muscaaurin, indicaxanthin, betanin, iso-betanin and betanidin [101].

In another study, 18 betacyanins and 23 betaxanthins were identified from Mexican cultivars “Roja Lisa” red, “Selection 2-1-62” yellow-orange and “Cristalina” green [102]. Recently 14 betalains, 9 betaxanthins and 5 betacyanins were identified from the Canary Island varieties “Colorada”, “Fresa”, “Blanco Buenavista” and “Blanco Fasnia” [41].

No general consensus exists regarding which part of the fruit has better yield for betalain extraction. Instead, it seems that the cultivar plays a key role in betalain extraction, e.g., yellow fruits are the most suitable for indicaxanthin extraction, whereas reddish fruits for would be more appropriate for betanin [7]. A summary of *O. ficus-indica* betalains quantification is presented in Table 4.

Since *O. ficus-indica* betalains are suitable for foodstuff coloring, several studies have been performed to evaluate the influence of pH, temperature, heating time, mass–solvent ratio for optimal pigment extraction [106] and clarification methods [107]. Yellow betaxanthin from Moroccan fruits showed thermal stability up to 70 °C for 30 min [100]. Betacyanin stability showed that temperatures above 70 °C reduced, in 75%, the stability of the pigments and betalains maintaining their integrity over a pH range from 3–7, with a maximum Chroma at 6.5 pH [99].

Moreover, pigment stability could be improved with the addition of 1% of citric acid [108] or ascorbic acid [109]. Storage and industrial processes include pasteurization, drying technique, fermentation and juice concentration may alter betalain stability, retention and yield. As an example, freeze-dried tissues showed a higher yield for pigment extraction than tissues treated with other drying techniques (oven, microwave and dehydrator) [35]. Juice stored at 10 °C and 95% of relative humidity resulted in betalain yield increment compared with room temperature storage [27] and pigment retention only decreased 5% when samples were stored for more than 28 days at 4 °C, protected from light [110]. 

Despite the presence of amino acids and reducing sugars, no Maillard browning was observed in concentrated juice suitable for coloring foodstuff, obtained at pilot-plant scale from typical operations for juice production [111,112] or storage [111]. Additionally, juice treated with high hydrostatic pressure HHP at 550 Mpa increased betalain concentrations compared with traditional heat treatment [113]. On the other hand, even though HPP increased pigment extraction yield, it may have reduced pigment stability by 25% [20]. Recently, a betanin extraction and purification process using a two-phase system based on tetrahydrofuran (THF) and sodium salts has been proposed, achieving 32.1 mg betanin /L [114]. 

#### 2.1.3. Biological Activities

Following the screening of the selected references, 42.2% of the analyzed information is related to *O. ficus-indica’s* biological activities (156 records) (Figure 2). Due to the elevated number of records, the nature of the information and to simplify the analysis and discussion of the information, the data were subdivided into biochemical assays and biological assays (Figure 5). 

As shown in previous sections, *O. ficus*-indica fruits contain a wide variety of chemical compounds, some of which are believed to be associated with different biological properties. For example, phenolic compounds present complex quantitative structure–activity relation (QSAR) properties [115], a situation that may explain the polyphenol’s ability to interact with electron oxidants, helping in the prevention of damaging free radical formation into biological systems [116]. In addition, phenolic compounds such as Isorhamnetin-3-O-glucoside, kaempferol 3-O-arabinoside and quercetin 3-O-rutinoside contain multiple polar functional groups that confer selective or unselective binding sites to biologically important molecules normally associated with health benefits [11,117].

Prickly pear is an important source of betalains, which, in addition to being of great interest as food colorings, could be a source of pharmacologically active compounds [118]. Compounds such as indicaxanthin and betanin could be important for maintaining a balance between free radicals and antioxidant defenses [80,119,120]. Additionally, there is evidence that betalains such as proline-betaxanthin, betanin, 17-hydroxy betanin and indicaxanthin can be used in the treatment of cancer [13,121,122].

It is important to highlight that most of the information on prickly pear fruit activities has been obtained from studies using plant extracts as raw material and less frequently from isolated or purified compounds. This gap makes it necessary to increase the number of studies related to the isolation and purification of target compounds in order to link the described effects to the responsible structures.

##### Biochemical Assays

From the biological activities records, 66.7% of the information corresponds to biochemical assays. This information was subdivided into different assay methods, taking into account the frequency of the mentioned methodologies (Figure 6).

The biochemical interactions of *O. ficus-indica* fruit compounds have been determined through numerous spectrophotometric methods. Most of the methodologies correspond indirectly to the ratio of electrons/hydrogens that a certain mixture or compound can shift with the oxidant probe or substrate [123]. Nowadays, these principles are widely used across several methodologies and their importance is reflected in the high number of published documents attempting to give insight into the beneficial effects of different food matrixes, vegetables, fruits, spices and herbs [124]. Thereby, it is extremely important to understand the conceptual and technical limitations [125] to increase the quality of the obtained information. Factors such as blanks, reagent and probe concentration, light, kinetics, stoichiometry, pH, temperature and incubation time are commonly neglected and exert great influence upon the assays. This causes difficulties in the standardization of the assays [124,126]. The lack of standardization and the different ways of expressing the results (molar equivalent of Trolox, ascorbic acid, gallic acid and other oxidant equivalents, or in terms of EC_50_), makes it difficult to obtain an accurate idea of the results and also limits the direct comparison of the data extracted from different documents.

Overall, prickly pear fruits exhibit a high antioxidant capacity against free radical species, including superoxide radicals, hydrogen peroxide, hydroxyl radicals and singlet oxygen [80,89], which converts them as an important innovative and upcoming product for human health prevention against oxidative stress-related diseases. Detailed information is summarized in Appendix B.

The antioxidant potential is often related to the ascorbic acid, α-tocopherol, glutathione, carotenoids and phenolic compounds present in different tissues [127] and can be attributed to the structure–active relationship (SAR) and the interactions with the probe [128]. For example, glycosylation of molecules increases polarity, turning the molecule more hydrophilic [129] and the DPPH radical is spontaneously formed in organic solutions. This organic environment obstructs hydrophilic molecules by donating a hydrogen or an electron, thus reducing DPPH molecule [130]. 

On the other hand, the ABTS+· radical proved to be ambivalent and it can be solubilized in aqueous and in organic media [131]. Therefore, an aqueous environment promotes the hydrogen or electron donation of glycosylated molecules, corroborated by an increase in the antioxidant potential of *O. ficus-indica* fruit.

##### Biological Assays

The remaining analyzed data allocated in the category “biological activities” was further subdivided taking into account the frequencies of the biological assays reported (Figure 7). From this, 19.7% was allocated into oxidative stress and 16.4% lipid metabolism, 16.4% antimicrobial, anti-inflammatory 12% were the other main categories of biological activity. Finally, individual records (n = 12) were found and to facilitate their analysis, they were grouped into a single category called “other” and represent 13.1% of the total information.

##### Oxidative Stress

Wistar rats submitted to ethanol-induced injury and with a juice intake of 40 mL/kg b.w of Tunisian purple-skinned fruit resulted in 3.9-rold reduced malondialdehyde (MDA), 4.33-fold reduced protein carbonyl (PC), 2.23-fold increased GSH levels and 2.57-fold increased plasma scavenging activity in erythrocytes [28]. It also increased the activity of hepatic antioxidant defenses such as superoxide dismutase (SOD), catalase (CAT) and glutathione (GSH) by 1.3-fold, 2.68-fold and 3.44-fold, respectively, 2.42-fold reduced alanine aminotransferase (ALT), 2-fold reduced aspartate transferase (AST), and 3.22-fold reduced gamma-glutamyl transferase (GGT) produced by ethanol-induced injury in Wistar rats [132]. Radiation-induced colitis in female Wistar rats was significantly ameliorated by pretreatment of 1 g/kg body weight of peel extract for 10 days. MDA levels decreased 1-fold and SOD increased 2.7-fold compared to irradiated rats [93]. 

Enzymatic levels (SOD, GPx, GRx) of high fat-fed rats after *O. ficus-indica* vinegar administration of 7 mL/kg for seven days were slightly modified compared with the control [133]. A commercial product designed by Cactinea^®^, composed of spray-dried juice extract, tested in rats at a daily intake of 240 mg/kg for seven days, increased, by 1.03-fold, blood globular levels of glutathione peroxidase [134]. Daily intake of 100 mg/kg b.w peel or fruit extract increased GSH, SOD and CAT and decreased MDA levels in adult male Sprague-Dawley rats with aluminum chloride-induced neurotoxicity [45].

The aqueous extract of combined pulp and seed recovered endoplasmic reticulum homeostasis via the unfolded protein response pathway in Drosophila melanogaster [135].

PC-12 neuronal cell line viability increased up to 73%, after H_2_O_2_ insult, at 100 µg/mL of ethyl acetate extract of Korean “Saboten” fruits [136]. Portuguese wild fruits juice extracts showed an anti-proliferative effect on the human colon cancer cell line HT29, an effect that may be attributed to an increment of 12 to 16% of ROS compared to the control [57]. Extracts from yellow, red and orange fruits at 50, 75 and 100 μg/mL decreased lymphocyte mortality treated with 100 μM of t-BOOH, by approximately 4.0 to 55%, respectively, for red cactus pulp and peels, and 5.0 to 31% for orange and yellow pulp and peel extracts compared with non-treated cells [104].

In addition, some tests have been performed in humans. When compared with a 75 mg/day vitamin C intake, the intake of 250 g/day of *O. ficus-indica fruit* decreased, by 4.2 and 1.74-fold, MDA and LDL hydro-peroxide, respectively, and increased, by 1.48-fold, the GSH:GSSG ratio in healthy humans [137]. Additionally, MDA, total cholesterol (TC), creatine kinase (CK), LH and HDL levels decreased after 150 mL/day juice supplementation during the yo-yo intermittent recovery test (YYIRT) [138]. In a separate study, the consumption of juice reduced hydroperoxides and increased the ability to restore redox balance caused by high intensity exercise in healthy active women [139].

##### Lipid Metabolism

Collected information suggests that fruit consumption may have a direct influence on liver metabolism, exerting hepatic protection through several metabolic modifications. For example, a juice intake of 40 mL/kg b.w of Tunisian purple-skinned fruit increased SOD 1.38-fold, CAT 1.4-fold, and ethanol-induced damage in rat erythrocytes 1.27-fold [132], whereas of ethanol-induced erythrocytes damage was reversed in 149.9% [28]. The same dosage increased the hepatic activity of SOD, CAT and GSH 1.32-fold, 2.68-fold and 3.44-fold, respectively, decreased total cholesterol (TC) 1.33-fold, triglycerides (TG) 1.81-fold and reduced lipid peroxidation in ethanol-induced injury in rats [140]. Moreover, fruit vinegar seems to reduce histopathological lesions, decrease liver damage and regulate lipid metabolism caused by oxidative stress in high fat-fed rats [141].

Glutamine pyruvic transaminase (GPT) and glutamic oxaloacetic transaminase decreased after 24, 48 and 72 h of 3 mL/day juice intake in the CCL4 liver degenerative process of rats [142]. Jordanian fruit juice at 2 mL/kg bd administration reduced liver damage induced by Cyclophosphamide (CP) in mice [143]. N-butanol fractioned flavonoids at 10 and 100 µg/mL protected Sprague-Dawley rat hepatocytes against alcoholic oxidative stress, increasing cell viability up to 40% [144]. 

Administration of 0.3% peel extract of the “Verde Villanueva” variety with a hypercholesterolemic diet was able to reduce hepatic cholesterol levels by 35% in hyperlipidemic hamsters [13]. Fruit ingestion of 250 g/day significantly increased 111In–LDL and 111In–HDL binding by human platelets. Specific binding sites on platelets can be upregulated by prickly pear consumption [145].

*O. ficus-indica* vinegar consumption of 7 mL/kg/d reduced plasma TC in 31%, TG in 53.7%, LDL in 82.14% and increased HDL in 20.5% on high fat-fed rats [133]. Methanolic extracts of Sicilian pulp fruit at variable concentrations (1–5 mg) inhibited lipid oxidation in red blood cells from healthy patients in a dose-dependent manner [80]. Fruit ingestion of 250 g/day significantly increased 111In–LDL and 111In–HDL binding by human platelets, reducing hypercholesterolemia [145].

##### Anti-Microbial

Fruit extracts exhibited different levels of antimicrobial activity against several strains [11,58,146,147]. The antimicrobial activity normally depends on the nature of the biological sample and extraction method, for example, oven-dried fruits showed better activity compared to other samples [148]. A summary of the antimicrobial potential (both against bacteria and fungi) of *O. ficus-indica* fruits is presented in (Table 5).

##### Anti-Inflammatory

Fruit extracts exhibited also anti-inflammatory effects both in animals and in humans. Intake of yellow fruit, 20 g f.w. fruit equivalent /kg, significantly decreased the inflammatory response to carrageenin-induced rat pleurisy, inhibiting the exudate level, number of migrated leukocytes and decreasing the release of pro-inflammatory mediators, such as prostaglandin E2 (PGE2), NADPH oxidases (Nox), interleukin 1-β (IL1-β) and tumor necrosis factor-α (TNF-α) in Wistar rats [153]. Obese rats fed with vinegar (7 mL/kg daily) reduced obesity-induced heart injury trough anti-inflammatory effects, characterized by decreased pro-inflammatory biomarkers and anti-adiposity mechanisms [141]. 

Korean fruit extract administration on female Sprague-Dawley rats and male IRC mice showed a significant reduction by 1.52-fold in leukocyte migration, 20% β-glucuronidase and 63% swelling percentage after carrageenan-induced paw edema and gastric injuries [154]. Adult male Sprague-Dawley rats with aluminum chloride-induced neurotoxicity treated with 100 mg/kg b.w peel or pulp extract decreased TNF-α and nuclear factor kappa B (NF-κβ) and decreased IL1-β levels [45]. Nitric oxide accumulation decreased approximately 5-fold in female Wistar rats’ radiation-induced colitis [93].

Consumption of 200 g of ”Surfarina” fruits twice a day in a dietary regimen decreased pro-inflammatory cytokines in healthy humans [155]. Daily ingestion of juice drastically decreased recalcitrant cutaneous sarcoidosis in a 50-year-old Caucasian female patient [156]. “Sanguigna” fruit juice fermentation with L. mesenteroides modified anti- and pro-inflammatory cytokines in Caco-2/TC7 cells [152]. Purple-skinned “Pelota” fruits treated with high hydrostatic pressure decreased hyaluronidase inhibition by approximately 20% compared with non-treated samples [64].

##### Cytotoxic

The cytotoxic activity of *O. ficus-indica* fruit has been tested in several cancer cellular lines. For example, Mexican fruit juice of the “Pelon” variety at 0.5% final concentration, during 48 h of incubation, decreased the cell viability of mammary (MCF-7), prostate (PC-3) and hepatic (HepG2) cancer cell lines, in 7.6, 5.1, and 12.6%, respectively [13]. Juice of fruits collected in different regions of Portugal showed an anti-proliferative effect at all tested concentrations (2–20%) in the human colon cancer cellular line HT29 after 96 h of incubation [57]. Aqueous extract from yellow fruits showed a 50% cell viability reduction at 400 mg fresh pulp equivalent/mL in human colon cancer cells [122]. Increasing concentrations of pulp and peel ethanolic extract from Egyptian fruits decreased cellular viability of liver (HepG2), colon (Caco-2) and mammary (MCF-7) cancer cell lines [157].

##### Neuroprotector

The acetone fraction of fruit peel extract at 400 µg/mL alleviated neurodegenerative effects in flies, exerting anti-amyloidogenic potential and mitigating the disruption of lipid membranes [158].

Korean juice extract at 50, 100 and 200 mg/kg b.w reduced ethanol-induced psychomotor alterations in Sprague-Dawley rats [159]. Lyophilized peel and fruit methanolic extract did not show acute oral toxicity at 2 g/kg b.w adult male Sprague-Dawley rats. Moreover, daily intake of peel or pulp extract at 100 mg/kg reduced the acetylcholinesterase (AChE) in 55.9% and 31.9%, respectively, compared to AlCl3 values [45].

Methanolic extracts of Korean fruits at 1 mg/mL reduced LDH release induced by N-methyl-d-aspartate (NMDA 25 µM), kainate (KA 30 µM) and oxygen–glucose deprivation (OGD 50 min) induced neuronal injury in cultured mouse cortical cells in approximately 59%, 37% and 75%, respectively [159]. Cell viability of neuronal PC-12 lines pre-incubated for 24 h with 100 µg/mL of ethyl acetate extract fraction was increased up to 73% after H_2_O_2_ µM insult [136].

##### Blood Sugar Regulation

Animal experiments showed the potential of fruit extracts in blood sugar regulation. Oral intake of 5.88 mg/kg of blended pad and fruit extract (75:25) increased plasma insulin levels and lowered blood glucose in Wistar rats [160]. Acetone peel fruit extract of 2 g/kg efficiently reduced blood glucose levels in healthy “Balb-C mice [39].

Additionally, a commercial formulation under Opundia™ designation reduced blood glucose levels by 7% and increased serum insulin concentration by 28% in healthy athletes [161]. Opundia™ capsule intake increased serum insulin and may facilitate blood glucose disposal after exercise in healthy men [162]. Additionally, the same formulation reduced blood glucose concentration in obese pre-diabetic adults [163].

##### Cardiovascular

A commercial formulation under patent no. W098/39019 corresponding to 100 g of fruit was tested on male athletes, which improved heart rate variability and autonomic nervous system activity, improving performance during exercise and reducing fatigue after exercise [164]. On the other hand, daily administration of 150 mL of juice improved diastolic and systolic blood pressure and maximal heart rate after yo-yo intermittent recovery test [165]. Ingestion of 150 mL of fruit juice reduced autonomic cardiac regulation in healthy men feed with high-fat formulations [166].

##### Antigenotoxic

DNA damage of HT29 cells induced by 100 µM of H_2_O_2_ and treated with 10% *v*/*v* Sicilian yellow fruit aqueous extract decreased by approximately 10% after 24 h [167]. Still, increasing concentrations of Korean fruits’ methanolic extract (12.5–100 µg/mL) reduced human peripheral lymphocytes by 200 µM of H_2_O_2_-induced damage in a dose-dependent manner [168].

##### Others

Daily oral intake of 3 mL/rat of Sicilian fruit juice revealed protective action against ethanol-induced ulcers in rats, avoiding ROS excessive accumulation and maintaining the mucosal stability [81]. The ingestion of commercial capsules containing fruit extract resulted in a significant reduction in nausea, anorexia and dry mouth in healthy subjects with alcohol hangover symptoms [169]. 

Lyophilized powder juice extracted from Korean fruits and maltodextrin at 800–1600 mg/kg significantly reduced stress-induced acute gastric lesions in rats, reducing gastric mucosal TNF-α and myeloperoxidase [170]. Ethyl acetate fruit extract administered intraperitoneally at 100 mg/kg in BALB/c mice increased the sleeping time after thiopental exposure 60 mg/kg compared to non-administered mice, inhibiting the effect on the CNS associated with the GABAergic system, thus, suggesting a sedative-hypnotic effect [171]. Freeze-dried fruit powder may be effective and a safe alternative active ingredient to hormone replacement treatment for the management of postmenopausal symptoms [172]. Ovariectomized rats treated with 500 mg/kg/d had modified hepatic gene expression of cytochrome P450 (CYP450) and glucuronosyltransferase UGT isoforms, suggesting that OFI consumption may be used to mitigate postmenopausal symptoms in women [173]. 

Additionally, 240 mg/mL methanolic extract from yellow “Surfarina” Sicilian fruits resuspended in PBS inhibit 80% of the spontaneous contractions of mouse ileum longitudinal muscle by interfering with pathways of intracellular Ca2+ release in the smooth muscle cells [174]. Administration of 5 mL/100 g b.w of 15% infusion of Sicilian fruit aqueous extract reduced plasma uric acid levels in male Wistar rats [81]. Increasing doses of green-white juice fruit and aqueous seed extract showed a laxative effect on gastrointestinal motility of constipated-after-loperamide-treatment and healthy rats. Furthermore, fruit ripening stage modified small intestinal mobility, laxative effect and gastric emptying [175,176].

Extracts of yellow and red Spanish fruits showed a significant increase in the average lifespan of Caenorhabditis elegans from 12.8 to 13.7 days for 0.1% *w*/*v* of red extract and 14.2 days for 0.5% *w*/*v* of yellow extract [158,177]. Sicilian pulp and peel extracts from yellow, red and orange fruits showed a clear angiogenesis effect in fertilized *Gallus gallus* eggs at 50 µg of extract/egg, orange peel extract being the more active extract, with 90.73% vessel growth inhibition and yellow pulp extract being less active, with 54.35% vessel growth inhibition [97].

Incubation of human skin fibroblast with 1 and 10 µg/mL of fermented juice for 72 h white Aureobasidium pullulans KCCM 12017 and Pichia jadinii KFCC 11487P may reduce photoaging by UV- damage, increasing transforming growth factor-beta (TGF-β) [26]. Crude methanolic extract of fruit peel showed a tyrosinase inhibitory effect of 72% and successfully inhibited 88% of lipoxygenase effects. In addition, 1% of peel extract incorporated into an o/w emulsion was shown to protect the formulation after in vitro irradiation of UVA [57].

## 3. Conclusions

Fruit consumption is no longer a personal choice based on taste; instead, it has become a health and life quality issue because of fruits’ nutritional content, amount of minerals, fibers, vitamins and bioactive compounds. The present review aimed to examine and summarize the scientific literature concerning antioxidant properties, biological activities, chemical composition and pigment information on the published data of *O. ficus-indica* fruits. 

As an emerging fruit, inclusion of *O. ficus-indica* in the daily diet is a challenge; however, reported in vitro and biological activities data suggest that periodic fruit consumption reduces the risk of developing chronic diseases. Furthermore, its protective effects are related to organic acids, phenolic compounds, lipid content and the synergistic interaction between them. In addition, fruits represent a promising source of natural red and yellow pigments (betaine and betaxanthin), especially peels that have high levels of pigments and may represent up to 45% of the fruit. 

Betalains are neither allergenic nor toxic, and aside from its high molar extinction coefficients, which makes them a good candidate to replace synthetic colorants in the food industry, they may have antioxidant properties, increasing the beneficial properties of the final product. Apart from the medicinal and food industry potential, *O. ficus-indica,* due to its CAM metabolism, can be used as an agricultural alternative for farmers in arid or semiarid regions and those facing the effects of climate change, providing not only fruits but succulent flowers and pads (cladodes), increasing the commercial value of the species.

## Figures and Tables

**Figure 1 plants-12-00543-f001:**
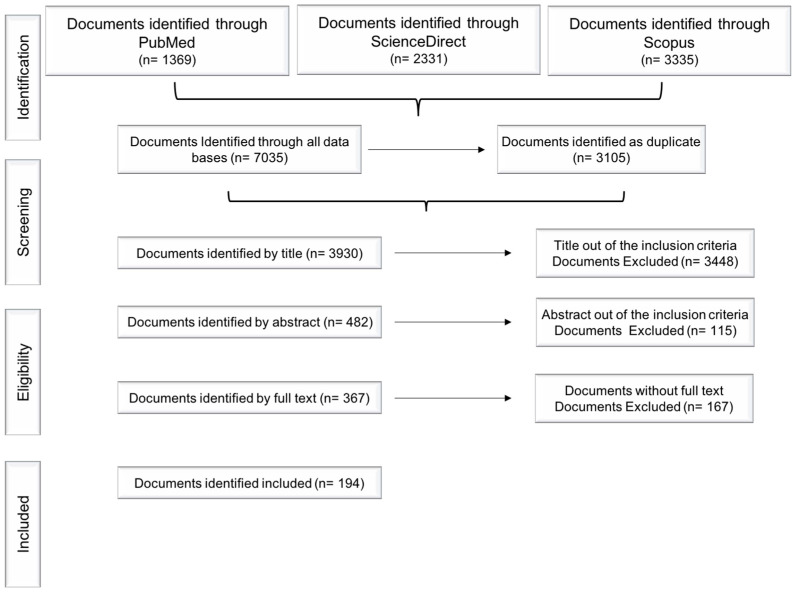
Flow diagram: search strategy and analyzed data.

**Figure 2 plants-12-00543-f002:**
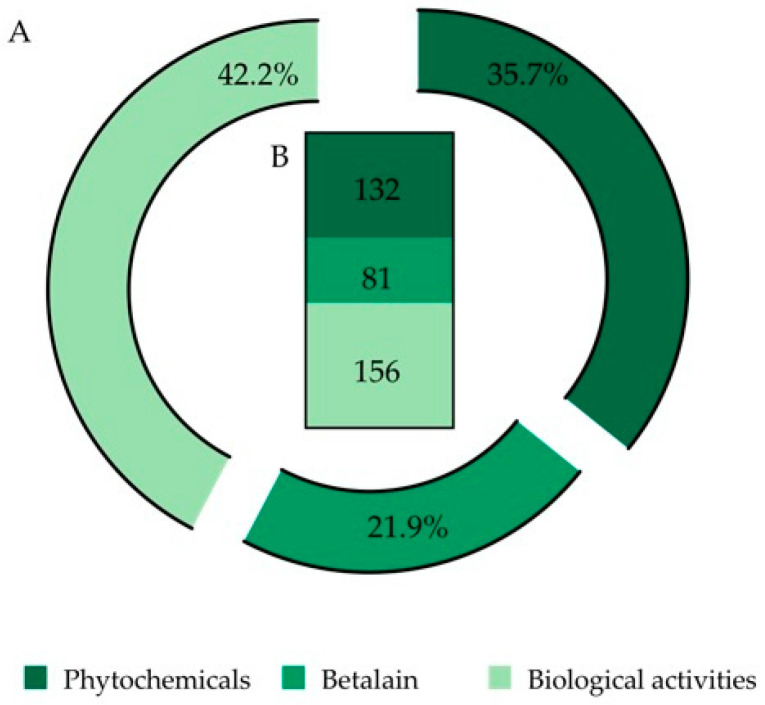
Final output of 369 records (from 194 selected papers) organized into three categories: phytochemicals, betalains and biological activities. (**A**) Data expressed in percentage. (**B**) Data expressed as number of records.

**Figure 3 plants-12-00543-f003:**
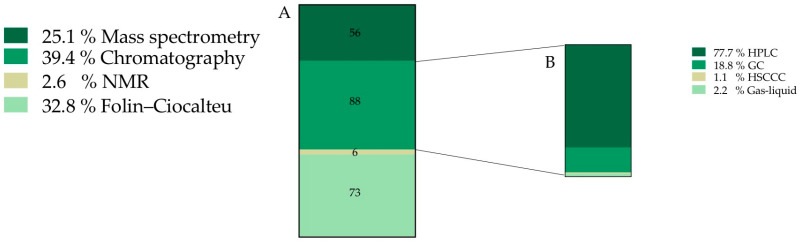
Frequency of records classified into chemical characterization category (expressed as percentage) (**A**) Techniques used to characterize and quantify *O. ficus-indica* chemical composition. (**B**) Type of chromatographic technique. **HPLC**—High performance liquid chromatography, **GC**—Gas chromatography, **HSCCC**—High speed counter current chromatography.

**Figure 4 plants-12-00543-f004:**
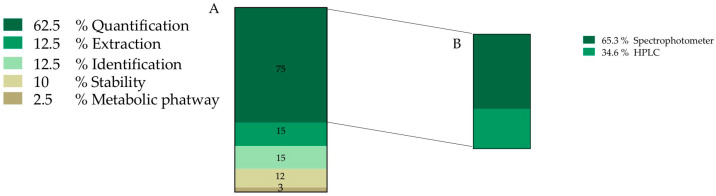
Frequency of records classified into pigment category (**A**) Data obtained from 120 records. (**B**) Methodology used for pigment quantification; **HPLC**—High performance liquid chromatography.

**Figure 5 plants-12-00543-f005:**
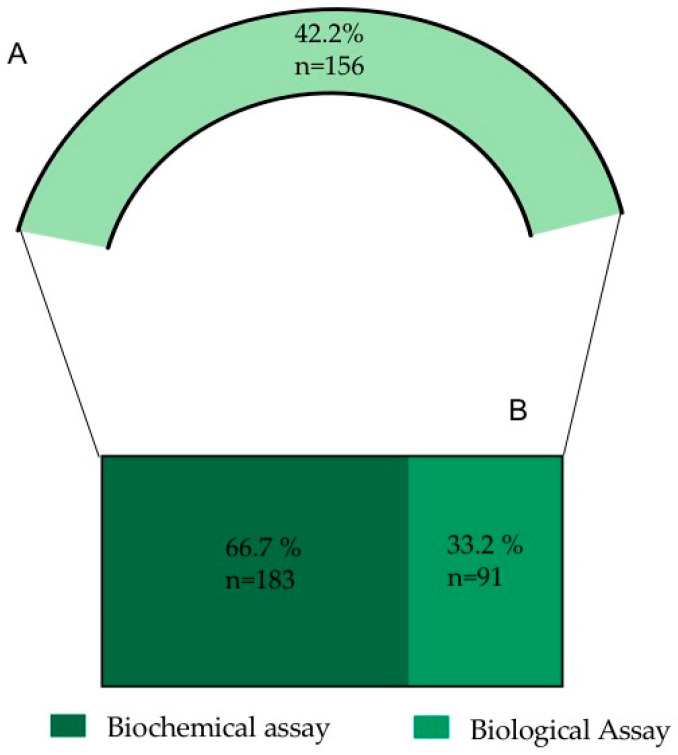
(**A**) Percentage of records dealing with biological activities (from a total of 396 records used in the study). (**B**) Resulting sub-categories after the biological activities analysis.

**Figure 6 plants-12-00543-f006:**
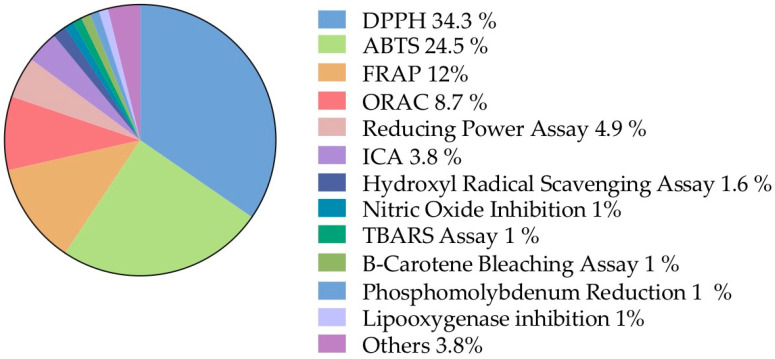
**Biochemical assays.** Information classified, according to the frequency of methodologies mentioned, into the biological activities sub-category. DPPH: 2,2-diphenyl-1-picrylhydrazyl; ABTS: green–blue stable radical cationic chromophore, 2,2-azinobis-(3-ethylbenzothiazoline-6-sulfonate); FRAP: ferric reducing antioxidant power; ORAC: oxygen radical absorbance capacity; ICA: iron chelating activity; TBARS: Thiobarbituric acid reactive substances.

**Figure 7 plants-12-00543-f007:**
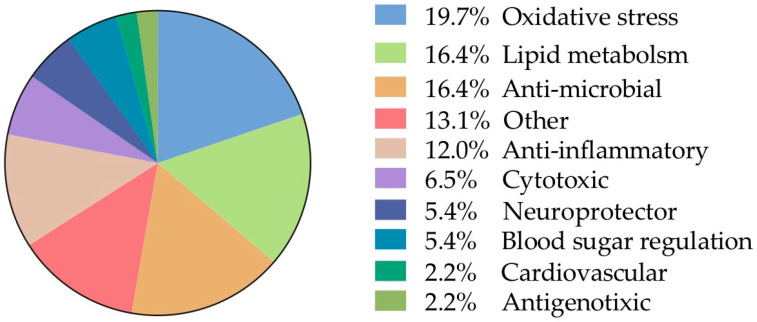
Frequency of sub-records classified according to the biological interaction (91 in total).

**Table 1 plants-12-00543-t001:** Total phenolic content of *O. ficus indica* fruits and juices.

Origin	Description	TP Values	Reference
**Algeria**	Fruit juice from different locations	493.5 to 618.5 GA mg/100 mL	[42]
**Algeria**	Orange fruit peel ethanolic extract	1.5 g GA /100 g d.w	[43]
**Algeria**	Fruit pulp extracted with different solvents	103.7 to 144.5 mg GA/100 g d.w	[44]
**Egypt**	Fruit methanolic extracts	Peel 165.2 mg GA/g d.w	[45]
		Pulp 53.8 mg GA/g d.w	
**Egypt**	Red-purple peel powder methanolic extract	52 g GA/100 g d.w	[46]
**Greek**	Fruit juice	9 mg GA mg/mL	[47]
**Italy**	Fruit juice	White “*Muscaredda*” 39.5 mg GA/100 g f.w	[48]
		Red “*Sanguigna*” 51.1 mg GA/100 g f.w	
		Yellow “*Sulfarina*” 45 mg GA/100 g f.w	
**Morocco**	Fruits from different locations	“*Amousten*” 33.7 to 44.7 mg GA/100 g f.w	[49]
**Morocco**	Fresh peel	“*Aakria*” 243.7 mg GA/100 g f.w	[50]
		“*Derbana*” 180 mg GA/100 g f.w	
		“*Safra*” 160 mg GA/100 g f.w	
**Morocco**	Pulp fruit acetone extract	“*Aakria*” 48 to 58 mg GA/100 g f.w	[51]
		“*Mlez*” 27 to 50 mg GA/ 100 g f.w	
**Mexico**	Fruit juice	Fruit juice 0.9 to 1 mg GA/100 g d.w	[52]
		Waste juice 0.6–0.8 GA/100 g d.w	
**Mexico**	Peel, pulp and juice	Red peel 722.5 mg GA/100 g d.w	[53]
		Red pulp 214.9 mg GA/100 g d.w	
		Red juice 209.2 mg GA/100 g d.w	
		White peel 748.6 mg GA/100 g d.w	
		White pulp 152.9 mg GA/100 g d.w	
		White juice 131.5 mg GA/100 g d.w	
		Yellow peel 444.6 mg GA/100 g d.w	
		Yellow pulp 168.3 mg GA/100 g d.w	
		Yellow juice 283.8 mg GA/100 g d.w	
**Mexico**	Raining season red fruit	230 to 450 mg GA/100 g d.w	[39]
**Mexico**	Fruit juice	Red juice 111.7 mg GA/100 mL	[54]
		Yellow juice 79.7 mg GA/100 mL	
**Peru**	Peel and seed	Purple peel 3.7 g GA/100 g d.w	[55]
		Purple seed 2.9 g GA 100 g/d.w	
		Yellow peel 3.4 g GA/100 g d.w	
		Yellow seed 2.7 g GA/100 g d.w	
**Peru**	Pulp ethanolic extract	693.2 mg GA/L	[56]
**Portugal**	Juice from different locations	2.1–2.5 g GA/L	[57]
**Portugal**	Yellow fruit	1.6 g GA/100 g d.w	[58]
**Saudi Arabia**	Peel and pulp	Peel 18.4 g GA/100 g f.w	[59]
		Pulp 9.7 g GA/100 g f.w	
**Tunisia**	Peel flour	2.7 g GA/100 g d.w	[60]
**Tunisia**	Spiny fruits	56 mg GA/100 g d.w	[6]
**Tunisia**	Whole fruit	1.5 g GA /100 g d.w	[6]
**Tunisia**	Red Peel & pulp	5 g GA/100 g d.w	[61]
		0.6 g GA/100 g d.w	
**Tunisia**	Peel, pulp and seed	Peel 4.7 g GA/100 g d.w	[62]
		Pulp 2.5 g GA/100 g d.w	
		Seed 108.6 mg GA/100 g d.w	

GA—gallic acid equivalents; d.w—dry weight; f.w—fresh weight.

**Table 2 plants-12-00543-t002:** Ascorbic acid concentration of *O. ficus indica* fruits and juices.

Origin	Description	Ascorbic Acid Concentration	Reference
**Algeria**	Homogenized orange pulp	24.34 mg/100 mL	[79]
**Algeria**	Orange skinned pulp juice	78.6 mg/100 mL	[42]
**Italy**	Sicilian white, yellow and red pulp	28; 29 & 30 mg/ 100 g f.w	[80]
**Italy**	95% red and 5% yellow blend fruit Juice	23.9 mg/100 mL	[81]
**Morocco**	Fruit juice	11.2 to 23.4 mg/L	[82]
**Morocco**	“*Aakria*” and “*Mles*” fresh peel	“Aakria” 70 mg/100 g f.w	[83]
		“Mles” 59 mg/100 g f.w	
**Morocco**	Oven dried peel	186 to 296 mg/100 g d.w	[84]
**Morocco**	“*Aakria*” and “*Mles*” pulp fruit	“Aakria” 26 mg/100 g f.w	[51]
		“Mles”24 mg f.w	
**Morocco**	“*Aakria*” and “*Safra*” oven dried and fresh peel	“Aakria” 295 mg/100 g d.w	[50]
		“Safra” 210 mg/10 g d.w	
		“Aakria” 74.7 mg/100 g f.w	
		“Safra” 57.2 mg/100 g f.w	
**Spain**	Red skinned blended fruit	18.5 mg/100 g f.w	[12]
**Spain**	Peel and Pulp	240 mg /100 g d.w peel	[85]
		140 mg /100 g d.w pulp	
**USA**	Texas Green skinned fruits	45.8 mg /100 g f.w	[14]
**USA**	Fruits growth on drainage sediment	54 mg/100 g f.w	[66]

**Table 4 plants-12-00543-t004:** Betalain quantification of *O. ficus indica* fruits and juices.

Origin	Description	Betaxanthin	Betacyanin	Betalain Content	Reference
**Algerian**	Homogenate pulp fruit juice	-	16.5 mg/100 mL	-	[42]
**Brazil**	Mesocarp ethanol/water 80:20 extract	14 mg/100 g f.w	3 mg/100 g f.w	15 mg/100 g f.w	[102]
	Endocarp ethanol/water 80:20 extract	15 mg/100 g f.w	2 mg/100 g f.w	16 mg/100 g f.w	
	Mesocarp + endocarp ethanol/water 80:20 extract	17 mg/100 g f.w	3 mg/100 g f.w	17 mg/100 g f.w	
**Italy**	Sicilian “*Gallia*” juice	4.8 mg/100 mL	0.6 mg/100 mL	-	[24]
	Sicilian “*Rossa*” juice	3.3 mg/100 mL	5.9 mg/100 mL		
**Italy**	Purple pulp and peel	-	39.3 mg/100 g f.w	-	[32]
**Italy**	Sicilian “*Agostani*” yellow juice	9 mg/100 g	-	-	[103]
	Sicilian “*Agostani*” red juice	8 mg/100 g			
**Italy**	Red cultivar	7.62 g /100 g d.w	33.59 g/100 g d.w	-	[104]
	Orange cultivar	20.26 g/100 g d.w	2.33 g/100 g d.w		
	Yellow cultivar	15.14 g/100 g d.w	1.61 g/100 g d.w		
**Italy**	Catania “*Agostani*” red fruit	2.6 mg/100 g f.w	4.8 mg/100 g f.w	-	[48]
	Catania “*Agostani*” Yellow fruit	6.8 mg/100 g f.w	1 mg/100 g f.w		
	Catania “*Agostani*” white fruit	0.3 mg/100 g f.w	0.2 mg/100 g f.w		
	Catania “*Bastardoni*” red fruit	2 mg/100 g f.w	2.8 mg/100 g f.w		
	Catania “*Bastardoni*” yellow fruit	5.1 mg/100 g f.w	0.6 mg/100 g f.w		
	Catania “*Bastardoni*” white fruit	0.3 mg/100 g f.w	0.3 mg/100 g f.w		
**Mexico**	Sanguinos pulp	0.9 mg/100 g f.w	2 mg/100 g f.w	-	[105]
	Sanguinos peel	0.6 mg/100 g f.w	5.7 mg/100 g f.w		
	Pelota pulp	3.9 mg/100 g f.w	27.9 mg/100 g f.w		
	Pelota peel	1 mg/100 g f.w	195. 6 mg/100 g f.w		
**Morocco**	“*Moussa*” yellow juice	37.83 mg/kg	-		[8]
	“*Moussa*” red juice	45.87 mg/kg			
**Morocco**	“*Amousten*” orange juice	87.7 μg/g	6.89 μ/g	96.59 μg/g	[49]
**Morocco**	“*Akria*” oven dried peel	-	-	35 mg/100 g d.w	[50]
	“*Safra*” oven dried peel			22 mg/100 g d.w	
	“*Akria*” fresh peel			6.2 mg/100 g f.w	
	“*Safra*” fresh peel			8.4 mg/100 g f.w	
**Spain**	Red-violet whole fruit water extract	-	15.2 mg/100 g f.w	-	[16]
	Red-violet whole fruit ethanol/water 80:20 extract		14.3 mg/100 g f.w		
	Red-violet whole fruit citrate-phosphate buffer pH 5.5 extract		14.5 mg/100 g f.w		
**Spain**	Whole red fruit extract	24.5 mg/100 g f.w	40.6 mg/100 g f.w	-	[12]
**Tunisia**	Peel fluor	250 mg/100 g d.w	336 mg/100 d.w	-	[60]
**Tunisia**	All fruit	843.6 mg/100 g d.w	1.4 mg/100 g d.w	-	[6]

**Table 5 plants-12-00543-t005:** Antimicrobial activity of *O. ficus indica* fruits.

Extracts	Strains	Anti-Microbial Activity	Reference
**Ethyl acetate extract**	*Leishmania mexicana*	243.9 µg/mL (IC_50_)	[149]
	*L. donovani*	70.3 µg/mL (IC_50_)	
**Aqueous extract of fruit syrup**	*Staphylococcus aureus* ATCC6538	665 µg/mL (MIC)	[11]
	*S. epidermis* CIP 106510	665 µg/mL (MIC)	
	*Bacillus cereus* ATCC1778	1.3 mg/mL (MIC)	
	*Escherichia coli* ATCC8739	1.3 mg/mL (MIC)	
	*Pseudomonas aeruginosa* ATCC9027	6.6 mg/mL (MIC)	
	*Salmonella sp*	6.6 mg/mL (MIC)	
	*Candida albicans* ATCC14053	6.6 mg/mL (MIC)	
**Hydro-alcoholic extract of peel**	*B. cereus*	75 µg/mL (MIC)	[150]
	*S. aureus*	150 µg/mL (MIC)	
	*Listeria monocytogenes*	150 µg/mL (MIC)	
	*Enterobacter cloacae*	75 µg/mL (MIC)	
	*P. aeruginosa*	150 µg/mL (MIC)	
	*Salmonella typhimurium*	150 µg/mL (MIC)	
	*Aspergillus fumigatus*	300 µg/mL (MIC)	
	*A. niger*	300 µg/mL (MIC)	
	*A. ochraceus*	100 µg/mL (MIC)	
	*A. versicolor*	300 µg/mL (MIC)	
	*Trichoderma viride*	150 µg/mL (MIC)	
	*Penicillium funiculosum*	150 µg/mL (MIC)	
	*P. ochrochloron*	75 µg/mL (MIC)	
	*P. verrucosum* var. cyclopium	300 µg/mL (MIC)	
**Extracts of fruits collected in summer**	*Mycobacterium tuberculosis* H37Rv	50 µg/mL (MIC)	[151]
**Extracts of fruits collected in rainy season**		100 µg/mL (MIC)	
**Aqueous isopropyl 80% peel extract**	*P. aeruginosa* ATCC9027	5.8 mm (IZ)	[46]
	*E. coli* ATCC11229	5 mm (IZ)	
	*S. aureus* NCTC10788	7 mm (IZ)	
	*S. typhi* ATCC14028	2.5 mm (IZ)	
	*F. culmorum* KF191	9.6 mm (IZ)	
	*F. culmorum* KF846	7.1 mm (IZ)	
	*F. graminearum* KF841	8.6 mm (IZ)	
	*F. oxysporum* ITEM 12591	6.4 mm (IZ)	
	*A. niger* ITEM 3856	7.8 mm (IZ)	
**Juice of from “*Bastardoni*” & “*Agostani*”**	*Salmonella enterica*	“Agostani” 0.6 cm (IZ)	[48]
		“Bastardoni” 0.55 cm (IZ)	
	*Pseudomonas fluorescens*	“Agostani” 0.23 cm (IZ)	
		“Bastardoni” 0.27 cm (IZ)	
	*E. coli*	“Agostani” 0.6 cm (IZ)	
		“Bastardoni” 0.55 cm (IZ)	
	*B. subtilis*	“Agostani” 0.77 cm (IZ)	
		“Bastardoni” 0.5 cm (IZ)	
**Fermented juice with *Leuconostoc mesenteroides***	*E. coli*	4.3 mm (IZ)	[152]
	*Bacilus megaterium* F6	4.2 mm (IZ)	
**Whole fruit ultrasound assisted extract**	Methicillin-resistant *Staphylococcus aureus* (MRSA)	27 mm (IZ)	[6]
	*B. cereus ATCC 14579*	14 mm (IZ)	
	*L. monocytogenes ATCC 19115*	18 mm (IZ)	
	*E. faecalis ATCC 29212*	10 mm (IZ)	
	*E. coli ATCC 25922*	15 mm (IZ)	
	*Klebsiella pneumoniae CIP 104727*	15 mm (IZ)	
	*Salmonella enteritidis DMB 560*	15 mm (IZ)	

## Data Availability

Not applicable.

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
