# Peer review of "Opuntia ficus-indica Fruit: A Systematic Review of Its Phytochemicals and Pharmacological Activities"

_plants, 2023, doi:10.3390/plants12030543_

Round 1
Reviewer 1 Report
att

Reviewer 2 Report
It is a high-quality review of the Opuntia-ficus indica plant's phytochemistry and biological-pharmacological properties. I have one major comment and some editing points to be amended for overall improvement.
1) The only thing I miss in this very good review is the connection (or at least an attempt) of the pharmacological properties with bioactive chemicals identified and reported. I recommend the authors to add a respective paragraph to show this connection with exemplary works (2-3 are enough).
2) Minor comments: Page 1 Abstract line 12 "intend", line 27, please add "Crassulaveau acid metabolism" before the abbreviation CAM. Page 2, lines 83-85 need to be rephrased. Page 3, line 105, records are 368 or 369? Page 6, line 172, delete "E.g." and begin with "For instance". Page 12, I see repeated paragraphs of page 10, please check and omit when required. Page 15, line 455 "tests". General comment, be consistent with "mL" in the text (capital L i mean).
Round 2
Reviewer 1 Report
Att.
